# Aberrant HMGA2 Expression Sustains Genome Instability That Promotes Metastasis and Therapeutic Resistance in Colorectal Cancer

**DOI:** 10.3390/cancers15061735

**Published:** 2023-03-13

**Authors:** Rubi Campos Gudiño, Kirk J. McManus, Sabine Hombach-Klonisch

**Affiliations:** 1Department of Biochemistry & Medical Genetics, Rady Faculty of Health Sciences, University of Manitoba, Winnipeg, MB R3E 0J9, Canada; 2CancerCare Manitoba Research Institute, CancerCare Manitoba, Winnipeg, MB R3E 0V9, Canada; 3Department of Human Anatomy and Cell Sciences, Max Rady College of Medicine, University of Manitoba, Winnipeg, MB R3E 0J9, Canada

**Keywords:** HMGA2, colorectal cancer, genome instability, metastasis, therapeutic resistance

## Abstract

**Simple Summary:**

Despite recent advances in early diagnostic and therapeutic strategies, colorectal cancer remains one of the most diagnosed and lethal cancers worldwide. Further, several patients are diagnosed at late stages with metastatic disease, ultimately limiting their therapeutic options. To better these statistics and overall patient survival, it is critical to identify and validate potential biomarkers (select genes) that contribute to colorectal cancer development (metastasis) and chemoresistance. Additionally, research has shown a clear association between genome instability, which confers an increased number of genetic alterations, and colorectal cancer. Although the aberrant expressions of various genes have been associated with early colorectal development, few have been validated as markers of metastatic disease. In this review, we highlight current research that suggests a member of the High-Mobility Group A gene family may be a potential biomarker of metastatic spread and therapeutic resistance in colorectal cancer. Further, we explain the mechanisms by which this gene, when aberrantly expressed, sustains genome instability in cancer cells that promote metastasis and chemoresistance in colorectal cancer, ultimately highlighting it as a promising biomarker that warrants further study.

**Abstract:**

Colorectal cancer (CRC) is one of the most lethal cancers worldwide, accounting for nearly ~10% of all cancer diagnoses and deaths. Current therapeutic approaches have considerably increased survival for patients diagnosed at early stages; however, ~20% of CRC patients are diagnosed with late-stage, metastatic CRC, where 5-year survival rates drop to 6–13% and treatment options are limited. Genome instability is an enabling hallmark of cancer that confers increased acquisition of genetic alterations, mutations, copy number variations and chromosomal rearrangements. In that regard, research has shown a clear association between genome instability and CRC, as the accumulation of aberrations in cancer-related genes provides subpopulations of cells with several advantages, such as increased proliferation rates, metastatic potential and therapeutic resistance. Although numerous genes have been associated with CRC, few have been validated as predictive biomarkers of metastasis or therapeutic resistance. A growing body of evidence suggests a member of the High-Mobility Group A (*HMGA*) gene family, *HMGA2*, is a potential biomarker of metastatic spread and therapeutic resistance. *HMGA2* is expressed in embryonic tissues and is frequently upregulated in aggressively growing cancers, including CRC. As an architectural, non-histone chromatin binding factor, it initiates chromatin decompaction to facilitate transcriptional regulation. HMGA2 maintains the capacity for stem cell renewal in embryonic and cancer tissues and is a known promoter of epithelial-to-mesenchymal transition in tumor cells. This review will focus on the known molecular mechanisms by which HMGA2 exerts genome protective functions that contribute to cancer cell survival and chemoresistance in CRC.

## 1. Introduction

### 1.1. Colorectal Cancer

Colorectal cancer (CRC) affects both males and females and is the third most commonly diagnosed and second most lethal cancer throughout the world. Every year, there are ~2 million new diagnoses and ~900,000 deaths, which account for ~10% of all cancer-related diagnoses and deaths, respectively [1,2]. Current therapeutic approaches, including endoscopic/surgical resection, radiotherapy and chemotherapy, have effectively improved overall survival rates; however, CRC incidence is still increasing and is predicted to reach over 3 million new diagnoses by 2040 [1,3,4]. Although the age-standardized mortality rate has declined and the current 5-year survival rates are ~65% [5,6], survival rates are still highly dependent on the stage at initial diagnosis. Statistics from the National Cancer Institute’s Surveillance, Epidemiology, and End Results (SEER) program show a 5-year survival rate of ~91% for the 37% of patients diagnosed with early, Stage I disease [7]. However, once the tumor has grown through the wall of the colon and spread to the lymph nodes, the 5-year survival rate decreases to ~72%, with 36% of patients being diagnosed at Stages II and III. Finally, once the cancer has metastasized to distant organs, the 5-year survival decreases to ~6–13%, with ~20% of patients diagnosed with advanced, Stage IV disease [7].

To reduce CRC morbidity and mortality rates, early screening modalities are critical to identify the existence of small, precursor lesions/tumors (polyps) and/or perform biopsies that can be used to provide a clear and early diagnosis [8,9]. Most screening programs employ fecal occult blood tests (FOBTs) as an initial non-invasive method; however, colonoscopies are considered the gold standard to detect the presence of pre-cancerous, benign polyps and/or potentially malignant lesions, and obtain tissue biopsies that will enable a more accurate diagnosis [10,11,12]. If a lesion/polyp is detected during a colonoscopy, surgical resection (polypectomy) is typically performed to evaluate the tumor through well-established histopathological analyses and can be further examined for critical molecular biomarkers that may assist in treatment decisions and management [12]. Overall, routine CRC screening has successfully reduced patient mortality by increasing the proportion of CRCs diagnosed at early stages, when the disease is more responsive to treatments, including surgery [7,10,11]. Unfortunately, as CRC is generally asymptomatic until late stages, ~20% of patients are diagnosed with advanced, metastatic disease [13], at which point the 5-year survival rate is as low as 6–13% and treatment options become more limited [14,15,16].

### 1.2. CRC Pathogenesis

Historically, CRC was considered as a homogeneous disease; however, more recent research studies have shown that there is a large degree of heterogeneity in the risk factors and mechanisms underlying CRC development and progression. Etiological factors include histology, molecular determinants (aberrant genes, proteins and pathways), region (proximal or distal colon) and inheritance (familial versus sporadic), as discussed below. CRC can be primarily divided into familial and sporadic forms, which account for ~20–25% and ~75–80% of CRC cases, respectively [17,18,19,20]. Lynch Syndrome, the most common hereditary form of CRC, confers a 75–80% lifetime risk of developing CRC (and extra-colonic cancers) relative to the ~7% lifetime risk for the general population [21,22,23,24,25,26]. Lynch Syndrome, which accounts for ~2–4% of all CRCs [23], is caused by inactivating germline mutations that typically occur in one of four DNA mismatch repair (MMR) genes (*MutL Homolog 1* [*MLH1*], *MutS Homolog 2* [*MSH2*], *MutS Homolog 6* [*MSH6*] and *Postmeiotic Segregation Increased 2* [*PMS2*]), or much rarer, *Epithelial Cell Adhesion Molecule* [*EPCAM*] alterations that adversely impact the *MSH2* promoter, preventing its expression [22,26]. Following the inheritance of an initial germline inactivating mutation, a subsequent somatic alteration inactivates the second wild-type allele (i.e., loss of heterozygosity) that inactivates the DNA mismatch repair pathway and underlies both microsatellite instability (MSI) and a hypermutation phenotype [27]. Although less common, Familial Adenomatous Polyposis CRC occurs in ~1% of all cases and arises due to defects in the *Adenomatous Polyposis Coli* (*APC*) tumor suppressor gene, which results in the aberrant activation of the WNT signaling pathway [20,28,29]. Similar to Lynch Syndrome, patients with a pre-existing germline *APC* mutation typically require a somatic alteration in the second allele for CRC to ultimately develop [30].

Conversely, sporadic CRC arises from the progressive accumulation of genetic and/or epigenetic alterations that provide selective growth and survival advantages to cells, which may cause them to transform into benign adenomatous polyps. In turn, these polyps may ultimately become malignant over an extended period of time (10 to 15 years [31]) due to the stepwise accumulation of subsequent genetic/epigenetic alterations in key genes that normally encode critical functions within cancer-promoting pathways, including cellular proliferation, cell cycle regulation, DNA damage repair/replication and apoptosis [20,32,33,34]. This temporal progression pattern was first described by Fearon and Vogelstein and is referred to as the “adenoma to carcinoma pathway” (reviewed in [20,33,35]), and it follows a relatively predictable series of genetic alterations; however, it is critical to highlight that not all CRCs follow this pattern or present all mutations described in this pathway. In general, the first step is proposed to involve the genetic inactivation/deletion of *APC*, which is followed by activation of the *Kirsten Rat Sarcoma Viral Proto-Oncogene* (*KRAS*) at the adenomatous stage. Finally, deletion of chromosome 18q and Tumor Protein 53 (*TP53*) inactivation occur during the transition into carcinoma [36]. However, not all sporadic CRCs harbor defects in *APC*, *KRAS* and/or *TP53*, and thus, additional genes are proposed to contribute to CRC pathogenesis, particularly those that impact genome stability and will adversely impact large cohorts of genes that may be critical for CRC pathogenesis [37,38,39,40]. Furthermore, recent evidence suggests a sub-set of sporadic CRCs may also arise from a characteristically different type of flat polyp known as a sessile serrated adenoma/polyp (SSA/P), which accounts for ~10–20% of CRCs [3,41]. The molecular determinants driving SSA/Ps appear to be distinct from those of conventional adenomatous polyps, which instead of following the “adenoma to carcinoma pathway”, are characterized by genetic/epigenetic changes in the B-Raf Proto-Oncogene, Serine/Threonine Kinase gene (*BRAF*) and a form of epigenetic instability more commonly referred to as the CpG island methylator phenotype (CIMP; reviewed in [42,43]), which is expected to promote the epigenetic silencing of numerous genes with pathogenic implications.

As a molecularly heterogenous disease, CRC can also be distinguished depending on its tissue of origin, that is, whether the primary tumor developed anywhere within the cecum, ascending or transverse colon (proximal or right-sided colon) versus the descending or sigmoid portion of the colon (distal or left-sided colon) or the rectum (Figure 1A) [44,45]. Clinical and genetic studies have established clear differences in the molecular determinants, histology, progression and survival between tumors arising in the different anatomical regions (reviewed in [3,45,46]). Many of the differences may arise due to the distinct embryologic origins, morphologies and microenvironments associated with proximal and distal tumors [45,46,47,48]. For example, the embryological cell of origin of the proximal colon is derived from the endoderm-derived midgut, whereas the distal colon and rectum arise from the hindgut [49,50,51]. These differences may also explain why SSA/Ps predominate in the proximal colon, while polypoid adenocarcinomas typically arise in the distal colon [45]. Additionally, the differences in biological (microbiome, lumen content, oxygen gradients and mucus production) and molecular features found throughout the colon and rectum also appear to be critical factors impacting disease pathogenesis and metastatic spread (Figure 1B) [45,46], as proximal cancers typically metastasize to the peritoneal cavity, whereas distal cancers commonly metastasize to the liver and lungs [45]. Accordingly, it is likely that a myriad of genetic, developmental, biological and environmental factors collectively contribute to the development and progression of CRC.

### 1.3. Genome Instability in CRC

Genome instability is an enabling hallmark of cancer [52,53] characterized by the accumulation of genetic alterations, including point mutations, insertions/deletions (indels), gene copy number alterations and chromosomal changes (e.g., translocations) [54,55]. Through genome instability, the alteration of cancer-associated genes (e.g., oncogenes and tumor suppressor genes) enables the acquisition of classic hallmarks of cancer. These hallmarks include uncontrolled proliferation and replicative potential, evasion of growth suppressors, resistance to apoptosis and increase of the invasiveness and metastatic potential of the tumor, which promote oncogenesis, disease progression and therapeutic resistance [52,53]. Genome instability is classically categorized into three distinct pathways: (1) MSI; (2) CIMP; and (3) chromosome instability (CIN) [55,56]. As stated above, MSI arises due to defects in DNA MMR genes, which underly a hypermutator phenotype that manifests as expansions or contractions of highly repetitive, microsatellite DNA [55,57,58]. CIMP is a DNA hypermethylation phenotype [59] that produces aberrant promoter hypermethylation and epigenetic silencing of genes, some of which may normally encode regulatory functions in key pathways, including DNA repair, DNA replication and cell cycle control [60]. Lastly, CIN is the most common and prevalent form of genome instability in CRC, as it is prevalent in ~85% of sporadic cases [54]. CIN is defined as an increase in the rate at which whole chromosomes, or large chromosome fragments, are gained or lost and is an ongoing driver of genetic and cellular heterogeneity [54,57,61]. Within this definition, CIN is divided into two main categories: (1) numerical CIN (N-CIN; gain/loss of chromosomes) and (2) structural CIN (S-CIN; translocation, amplification and deletions of large chromosome fragments) [62]. Conceptually, gains and/or losses of whole chromosomes and chromosome fragments impacts gene copy numbers and expression patterns, such that gains may correspond with the overexpression of oncogenes and losses with reduced expression of tumor suppressor genes [63,64,65]. CIN frequently arises due to defects in DNA replication, DNA repair, cell cycle regulation and/or defects that adversely impact chromosome dynamics during mitosis, including aberrant regulation of centrosome biology [66], chromosome cohesion or condensation [37], kinetochore function or microtubule dynamics [67] that adversely impact mitotic fidelity and genome stability. Based on the perpetual defects associated with these critical pathways, CIN is considered a dynamic phenotype, with gains and/or losses of specific chromosomes appearing to be largely random; however, more recent genetic evidence suggests certain chromosomes may be preferentially missegregated under specific conditions [68,69]. Furthermore, distinguishing between numerical and structural CIN may provide critical insight into the underlying defects giving rise to CIN; for example, segregation errors are typically associated with numerical CIN, whereas genotoxic stress would lead to structural CIN (reviewed in [62,70]. In any case, genome instability (MSI, CIMP and CIN) is heavily associated with CRC and is, therefore, proposed to be an essential driver of disease pathogenesis. Moreover, this instability is proposed to provide subpopulations of cells with selective growth advantages and disadvantages that, through selective pressures, are predicted to increase the likelihood of acquiring additional hallmarks of cancer, such as proliferative and invasive potential, to endow tumors with the ability to rapidly adapt to novel environments, develop therapeutic resistance and/or enhance metastatic capabilities [65,71,72,73,74,75].

While the “adenoma to carcinoma pathway” explains the stepwise gain of genetic mutations leading to CRC [20], the discovery of the two-phased cancer evolution model (or stochastic progression model) by Heng et al. [62] proposes a novel conceptual basis explaining the genetic mutations and karyotypic shuffling that contributes to genome instability and CRC development. The two-phase model is described by three key events: (1) stress-induced genome chaos; (2) selection of surviving, newly arranged genomes (i.e., macroevolution); and (3) gain of mutations in individual genes (i.e., microevolution). Briefly, stress-induced genome chaos enables a macroevolutionary event that encourages gross genomic rearrangements where a few stable cells with new genomes survive. This is followed by a microevolutionary event, in which surviving cells gain select mutations that ensure their adaptation and proliferation (reviewed in [62]), thus highlighting a potential method by which genome instability occurs and is subsequently maintained. Supporting the two-phased cancer evolution model, the “Big Bang” model (characterized in CRC) focuses on the continuous intra-tumoral heterogeneity that results from a single clonal expansion followed by numerous heterogenous subclones [76]. Following these events, select advantageous mutations may occur in different locations of the tumor that, although non-dominant, unlike “early” alterations, remain pervasive and potentially result in aggressive sub-clones. Ultimately, these virtually undetectable and highly heterogeneous subclones may provide the resistant characteristic that is well established in metastatic CRCs [76].

### 1.4. HMGA2 and CRC

Our current knowledge highlights that metastatic organotropism (i.e., non-random distribution of metastases) is regulated by several factors, including the molecular features of cancer cells and the tumor microenvironment [45,47,48]. Evidence shows numerous genes are implicated in CRC pathogenesis and disease progression; however, few of these have been successfully validated as predictive biomarkers for metastases and therapeutic response. One potential biomarker of metastatic spread and therapeutic resistance is a member of the High-mobility Group A (*HMGA*) gene family, *HMGA2* (or *HMGI-C*), characterized by its functional motif containing three DNA-binding “AT hooks” encoded by the first three exons (Figure 2) and known to bind to Adenine-Thymine (AT)-rich DNA sequences [77,78].

*HMGA2* localizes to 12q14.3 in humans and contains five exons [79]. The encoded protein (Figure 2) contains three positively charged basic AT hook domains that are predominantly comprised of highly conserved palindromic aa sequences enriched for Arginine and Lysine residues that facilitate binding to AT-rich sequences in the minor groove of B-DNA [80]. The spacing between the three AT hooks is determined by linkers 1 and 2 (Figure 2A), while linker 3 connects the acidic carboxy (C-) terminus [80,81]. Key structural differences between HMGA2 and the related protein HMGA1, such as aa composition, length of the linker regions and a two aa difference (Lysine/Arginine) in the third AT hook, may help explain the flexibility of the molecule [81] and the distinct biological functions ascribed to HMGA1 and HMGA2 [80,81], as evidenced by the disparate phenotypes in genetic knockout mice [82,83,84]. For example, *Hmga1* deficiency leads to cardiac hypertrophy, diabetes and lymphoid malignancies in mice [82,83], whereas *Hmga2* knockout mice exhibit reduced growth (“pygmy” phenotype), diminished adipose tissue and male infertility [84,85]. AT hook 2 also determines nuclear localization of HMGA2 through binding to the nuclear import protein importin-alpha2 (IMPA2) [86]. The C-terminus of HMGA2 is the substrate for many post-translational modifications (PTMs), which influence its functions by regulating DNA-binding and protein interactions [86,87]. Additionally, the aa sequence differences that occur within the C-terminal regions of HMGA1 and HMGA2 also explain the differential PTMs that can occur for both of these proteins [88], with HMGA2 (K_90_PAQEE**T**EE**TSS**QE**S**AEED_108_) having five S/T residues (bold) capable of being phosphorylated by Casein Kinase 2 (CK2), compared to only three in HMGA1 [88]. Furthermore, Protein Kinase C (PKC), Cyclin Dependent Kinase 1 (CDK1) and Ataxia Telangiectasia Mutated (ATM) all phosphorylate distinct domains of HMGA1, resulting in reduced protein binding ability [89,90]. Phosphorylation at the Serine-Glutamine (SQ) motif by ATM in the C-terminus has been described for HMGA1 [91], while HMGA2 is also a target for ATM kinase activity [92]. HMGA2 is a substrate for additional PTMs, including poly-ADP ribosylation (or PARylation); however, the location and the impact PARylation has on HMGA2 function remain unknown [93]. Deletion of the HMGA2 C-terminus alters its DNA and protein binding functions [87,90], indicating that the C-terminal domain serves an important role in negatively modulating HMGA2 interactions. Additionally, AT hook 2, in conjunction with the adjacent aa sequences of linker 2, have been identified as the essential protein–protein interacting domain of HMGA2 [90] (Figure 2). Thus, HMGA2 is heavily regulated by PTMs that will ultimately define its functions in maintaining genome stability. Known HMGA2 variants that may affect its PTMs and function are summarized in Table 1.

### 1.5. The Role of HMGA2 in Genome Maintenance

HMGA2 assumes its role as an architectural transcription factor [97] in chromatin remodeling to promote very diverse biological functions involving protein binding, stabilization of distinct DNA conformations and transcriptional regulation. HMGA2 competes with the H1 linker histone for binding to DNA [98], resulting in interphase chromatin decompaction [81,99] and enhanced access for transcription factors. This function is especially important in cells with high plasticity, such as embryonic stem cells and cancer cells, both of which display high levels of HMGA2 [100].

Although the AT hook domains bind to AT-rich sequences in B-DNA, HMGA2 also has sequence-independent high-affinity binding for distinct DNA conformations, such as branched structures including three- and four-way junctions [99,101], along with bent and supercoiled DNA [102,103]. These non-B-form DNA conformations are transient in nature and occur frequently in replicating cells [104]. In cancer, high-fidelity DNA replication is an essential requirement for tumor cells with high proliferative capacity, as DNA base lesions or single-strand DNA breaks can cause replication fork stalling that may eventually lead to lethal double-strand breaks (DSBs). In 2014, Yu and colleagues [101] demonstrated that HMGA2 binds with high affinity to forked DNA structures in vitro and associates with stalled replication forks in cells. Moreover, they determined that HMGA2 co-localized with the replication fork proteins, Proliferating Cell Nuclear Antigen (PCNA) and Replication Protein A (RPA), at sites of ongoing and arrested DNA replication forks, which was associated with reduced DSBs following hydroxyurea (HU)-induced replication fork stalling in both mouse embryonic fibroblasts and human cancer cells. Furthermore, through its interaction with the replication fork proteins PCNA and RPA, they showed that HMGA2 stabilizes arrested replication forks to protect them from endonucleolytic attack. Importantly, all three AT hooks were required for effective protection of these stalled replication forks [101], supporting a model in which HMGA2 AT hooks bind to different DNA segments [81] and, thus, HMGA2 serves as a scaffolding protein to stabilize forked DNA conformations.

Beyond binding to distinct DNA conformations, HMGA2 also participates in base excision repair (BER), as the three AT hooks appear to possess enzymatic activity and function as apurinic (AP) and deoxyribose phosphate (dRP) lyases [105]. The ability of HMGA2 to enhance the cellular capacity for cleavage of AP/dRP sites removes the processing bottleneck in BER. With an estimated 70,000 base lesions occurring every day per mammalian cell [106], this DNA damage repair mechanism is not only crucial for stem cells, but also for highly proliferative neoplastic cells, as unresolved base lesions lead to replication fork stalling and high risks for DSBs following replication fork collapse. The BER-supporting function of HMGA2 enhances the ability of tumor cells to promote the timely repair of base lesions, thus reducing replication fork stalling and promoting genome maintenance under DNA stress. Similarly, alkylating DNA damage also requires BER to replace alkylated bases. Additionally, HMGA2 protein overabundance sustained both Ataxia Telangiectasia and Rad3-related (ATR) protein activity and Checkpoint Kinase 1 (CHEK1) DNA damage signaling following alkylating DNA damage associated with methyl methanesulfonate (MMS) treatment. Moreover, increased HMGA2 abundance was also associated with a decrease in DSBs in cancer cells and causally linked to increased CHEK1 phosphorylation, prolonged G2/M arrest and reduced apoptosis [107,108], suggesting that HMGA2 facilitates expeditious DNA repair to prevent DSBs and subsequent apoptosis induction.

Telomere instability is another hallmark of cancer progression [109,110], and many cancers exhibit increased telomerase activity to maintain minimal telomere length over numerous cycles of mitotic DNA replication [111]. Enhanced transcriptional regulation of human Telomerase Reverse Transcriptase (hTERT; catalytic subunit of telomerase) was reported upon HMGA2-Specificity Protein 1 (SP1) interaction at the hTERT promoter in HepG2 cells [112]. Independent of telomerase activity and in interphase nuclei, HMGA2 binds to Telomeric Repeat Binding Factor 2 (TRF2) [113], the central hub protein of the Shelterin complex that caps chromosomal ends to prevent them from being recognized as DSBs [114]. The first linker domain and the AT hook 2 domain of HMGA2 were deemed essential for the TRF2 interaction, and this interaction increased TRF2-binding to telomeric DNA that enhanced telomere stability, as demonstrated by a reduction in the number of TIFs (telomere dysfunction induced foci) [113]. Moreover, *HMGA2* silencing corresponded with an increase in telomere aggregates, anaphase chromatin bridges, micronuclei formation and signs of telomere dysfunction, which culminated in chromosome segregation defects and increases in aneuploidy in cancer cells [113]. Thus, these observations highlight the critical requirement for *HMGA2* expression to maintain chromosomal telomeres and ultimately sustain and promote viable cell divisions of cancer cells.

HMGA2 also appears to exhibit a critical role in genome maintenance based on its ability to enhance PARP1 (Poly [ADP-ribose] polymerase (1) activity. In this regard, HMGA2 was shown to physically interact with PARP1 [93], which stimulates PARP1 enzymatic activity on alkylated DNA from cancer cells treated with MMS. High cellular HMGA2 protein levels were associated with earlier and increased DNA damage-induced PARylation activity by PARP1, and consequently, higher concentrations of PARP inhibitors were required to inhibit PARP1 activity. Importantly, HMGA2 was able to mitigate PARP inhibitor-mediated trapping of PARP1 to DNA, resulting in reduced DSBs, as determined by reduced γH2AX foci (surrogate marker for DSBs), along with apoptosis inhibition. Furthermore, the anti-apoptotic and PARylation-inducing effects of HMGA2 were dependent on functional AT hooks [93]. Through its combined AP/dRP lyase activity and its PARP1-promoting function, HMGA2 supports cellular capacity for BER and single-stranded (ss) DNA repair, which are both particularly important in cancer cells undergoing oxidative DNA damage or those exposed to alkylating drugs.

Proliferating cells utilize Topoisomerase (Topo) I and II enzymes to relax supercoiled DNA structures at advancing replication forks, and both enzymes transiently induce controlled DNA breaks [115]. HMGA2 binds Topo I [116] and enhances Topo I and II functions [102,116]. Importantly, HMGA2 stabilizes supercoiled DNA in front of replication forks [117], which is facilitated by the linker spacing between the three DNA-binding AT hooks [81]. Binding of HMGA2 to adjacent DNA segments at supercoiled DNA [118] requires more than one AT hook, leads to gradual relaxation of supercoiled DNA structures by Topo I and aids in stabilization of tertiary structures at the replication forks [118]. This HMGA2 protective effect is particularly evident at subtelomeric regions, which are vulnerable during DNA replication due to DNA supercoiling in the vicinity of complex telomeric structures [103]. Supercoiled DNA, ssDNA and forked DNA structures are all present at stalled replication forks, and HMGA2 binds to each of these DNA conformations with higher affinities than to canonical B-form DNA [101,118]. The above-mentioned mechanisms detail how HMGA2 reduces DSBs and prevents cell cycle arrest, which highlights the important protective role HMGA2 has during replication stress in highly proliferative cancer cells.

Knowledge is currently lacking on the potential role of HMGA2 in DSB repair. In two CRC cell lines (HCT116 and SW480) with plasmid-driven *HMGA2* overexpression, γH2AX foci were present at several time points following the irradiation of cells [119], which is consistent with HMGA2 inhibiting non-homologous end joining (NHEJ) and DSB repair. This observation contrasts with that of HMGA1, which appears to promote NHEJ by increasing DNA Ligase IV activity [120]. Interestingly, HMGA2-mediated increased transcriptional ATM gene activity is enhanced with DSB-induced ATM kinase activity [92]. HMGA2 not only binds ATM, but is an ATM substrate upon DNA damage [92], suggesting that HMGA2 functions in DSB signaling and repair upon genotoxic stress in cancer cells. Impaired DNA damage repair following DSB induction in cells with reduced HMGA2 protein levels may in part be caused by reduced ATM protein levels [92]. Although increased DNA Ligase IV activity and a supportive function in NHEJ was identified for HMGA1 [120], the potential role of HMGA2 in DSB repair remains controversial. A delay of DNA-PKcs release from DSBs with consecutive inhibition of NHEJ by HMGA2 was observed [121], although this was shown in model systems with exogenous *HMGA2* overexpression. The HMGA2 function in supporting genome maintenance explains its mitigating effects on treatment efficacy in CRC, which will be described below (Figure 3).

### 1.6. Aberrant HMGA2 Expression Promotes CRC Metastasis

As indicated above, *HMGA2* maps to chromosome 12q14.3 and encodes a small (109 amino acid [aa]) non-histone, chromatin-associated protein. HMGA2 modulates transcription of target genes through the alteration of chromatin architecture, thus affecting a variety of downstream biological processes [119]. During transcription, and depending on the number and spatial proximity of AT-rich binding sites, HMGA2 can either enhance or repress transcription of key genes [122]. In a cancer context, increased *HMGA2* expression in CRC corresponds with enhanced cellular growth, differentiation, proliferation, transformation, angiogenesis, epithelial-to-mesenchymal transition, apoptosis and metastasis [117,119,123,124].

Growing evidence supports the role of increased *HMGA2* expression in CRC progression and metastasis, which suggests *HMGA2* may be a prognostic marker of poor patient prognosis, as it is a critical regulator of various signaling pathways [119]. Indeed, previous studies show increased *HMGA2* expression may promote CRC metastasis by enhancing P53 ubiquitination and its subsequent degradation through the E3 ubiquitin ligase MDM2, transcription regulation of *FN1* and *IL11* (i.e., *IL11/STAT3* pathway) [125] and activation of the WNT/ß-catenin pathway [126]. In addition to signaling pathways, non-coding RNAs (ncRNAs), including microRNAs (miRNAs), long non-coding RNAs (lncRNAs) and circularRNAs (circRNAs), have been identified as either targets or promoters of *HMGA2* expression in a CRC specific context (Table 2), in which aberrant activity frequently correlates with more aggressive CRCs (i.e., metastasis and therapeutic resistance). As such, studying *HMGA2* as a predictive biomarker for metastasis and treatment resistance in CRC is critical to obtain further insight into its pathogenic properties and its roles in sustaining genome instability, disease progression and metastasis, as it may ultimately serve as a target for personalized therapeutic intervention.

### 1.7. HMGA2 Promotes Chemoresistance in CRC

Treatment options for patients with advanced and metastatic CRC currently include surgery, radiation therapy and chemotherapy, with few targeted therapy options to curb angiogenesis and growth factor signaling. Approved chemotherapeutic compounds for CRC are 5-fluorouracil (5-FU), Capecitabine, Oxaliplatin and Irinotecan [139]. For many years, the standard treatment for CRC has been the antimetabolite 5-FU, but treatment resistance leads to treatment failures negatively affecting patient outcomes [140]. In practice, 5-FU causes base lesions that promote replication fork stalling in proliferating cancer cells, and *HMGA2* overexpression is associated with 5-FU chemoresistance in CRC patients [141]. By enhancing PARP1 activity [93], increasing the BER capacity in cells [105] and stabilizing stalled replication forks [101], HMGA2 limits the efficacy of 5-FU and Capecitabine in CRC patients with *HMGA2*-expressing tumors. Additionally, the PARP inhibitor (PARPi) Olaparib is currently being evaluated in a Phase II clinical trial (PEMBROLA) for CRC patients. HMGA2 was shown to enhance PARP1 enzymatic activity and prevent PARPi trapping to DNA [93], which is considered the main mechanisms of their cytotoxicity, and thus antagonizes PARPi efficacy.

During DNA replication, DNA supercoiling in front of the replication fork creates tensions in the superhelix that are released by Topo I and II enzymes through transiently creating DNA strand breaks [142]. Clinically used Topoisomerase inhibitors, such as Camptothecin and Irinotecan, maintain these enzymes covalently bound to DNA strand breaks, effectively trapping Topo I and II, which in turn promotes replication fork stalling and collapse, leading to cytotoxic DSBs [143]. As indicated above, HMGA2 enhances the enzymatic activities of Topo I and II [102,116] and HMGA2 abundance-determined sensitivity to SN38, the active form of Irinotecan [103]. In line with the cell-based findings, patient-derived CRC xenograft tumors with low to moderate HMGA2 levels were deemed more sensitive to Irinotecan [103]. HMGA2 binding to supercoiled DNA structures may interfere with the Topo-I/DNA complex formation in a concentration-dependent manner [103], highlighting that HMGA2 abundance in individual CRC tumors may determine sensitivity to Topo I and II inhibitors.

Finally, Oxaliplatin induces intra-strand dinucleotide DNA adducts that are repaired by nucleotide excision repair and are frequently used to treat metastatic CRC [144]. Recently, the long non-coding RNA PiHL (P53 inHibiting LncRNA) was identified to induce Oxaliplatin resistance in CRC through reactivation of *HMGA2*. PiHL reduces (Enhancer of Zeste Homolog 2) EZH2-mediated tri-methylation of histone H3 at lysine 27 (H3K27me3) at the *HMGA2* promoter to de-repress *HMGA2* expression in CRC (Table 2) [137]. Moreover, PiHL knockdown reduced HMGA2 levels and restored sensitivity to Oxaliplatin treatment in a CRC xenograft model [137], thus linking HMGA2 tumor levels to Oxaliplatin resistance in CRC.

## 2. Conclusions

HMGA2 cellular functions in chromatin remodeling and genome maintenance depend on the cellular context, protein abundance, specific biological process and the functionality of other DNA maintenance/repair factors. HMGA2 may prevail, resulting from stress-induced genome instability during the macroevolution stage of CRC evolution. Select subclones may subsequently exploit HMGA2 critical functions in genome maintenance, apoptosis prevention and regulation of downstream signaling pathways to sustain genome instability in CRC (Figure 3). Thus, HMGA2 facilitates adaptations in the CRC subclonal microevolution of CRC tumors that promote growth in the metastatic niche and survival under therapeutic pressure. This strongly supports the clinical relevance of HMGA2 as a potential novel predictive biomarker of metastasis and therapeutic resistance in CRC, which also warrants further study as a potential target for personalized therapeutic intervention.

## Figures and Tables

**Figure 1 cancers-15-01735-f001:**
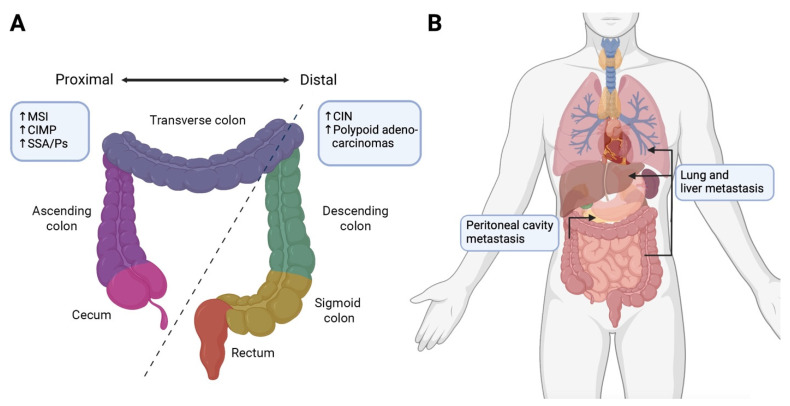
Anatomy of the colon and rectum and predominance of metastatic sites relative to proximal and distal CRCs. (**A**) Prevalence of distinct features varying between proximal and distal CRCs. (**B**) Predominant metastatic sites according to CRC sidedness: proximal CRCs majorly metastasize to the peritoneal cavity, whereas distal CRCs metastasize to the lungs and the liver. MSI, microsatellite instability; CIMP, CpG island methylator phenotype; CIN, chromosome instability; SSA/P, sessile serrated adenoma/polyp.

**Figure 2 cancers-15-01735-f002:**
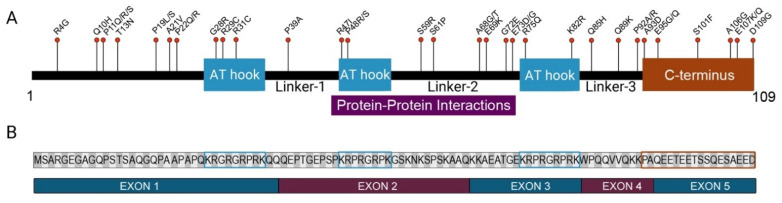
Distribution of encoded alterations of HMGA2. (**A**) Schematic depicting the amino acid position (red) of encoded alterations that are predicted to be possibly/probably damaging (PolyPhen-2) or deleterious (SIFT). Amino acids are indicated by their single letter code, numbers denote the amino acid position in the HMGA2 protein. (**B**) Distribution of the complete amino acid sequence (top) and exon positions (bottom) of the HMGA2 protein. AT hooks amino acid sequences (KRGRGRPRK, KRPRGRPK and KRPRGRPRK, respectively) are highlighted in blue and the C-terminus amino acid sequence (PAQEETEETSSQESAEED) in orange.

**Figure 3 cancers-15-01735-f003:**
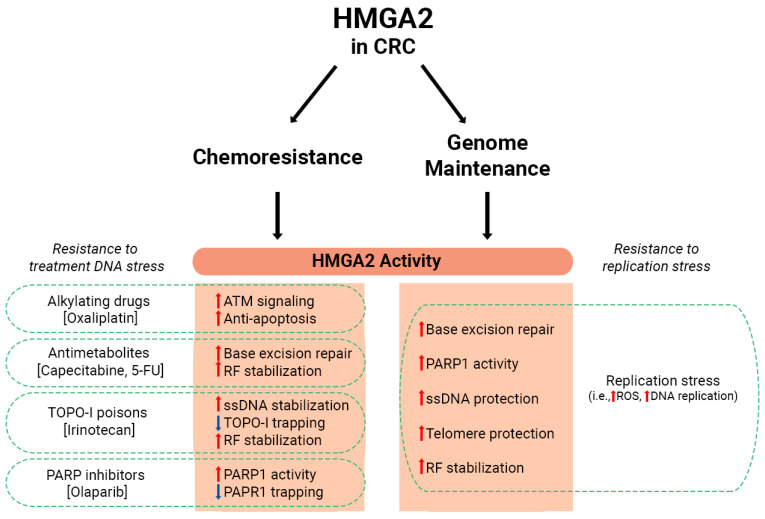
HMGA2 supports chemoresistance and genome maintenance in CRC. Schematic depicting the mechanisms by which HMGA2 counteracts treatment-induced DNA stress and protects genome integrity. RF, replication fork; ssDNA, single strand DNA; ATM, ataxia telangiectasia mutated; ROS, reactive oxygen species; PARP, Poly [ADP-ribose] polymerase 1; TOPO, topoisomerase.

**Table 1 cancers-15-01735-t001:** HMGA2 missense variants [94] with predicted consequence and clinical significance of amino acid substitution by PolyPhen-2 [95] and SIFT [96].

Variant ID	Amino Acid Position ^A^	Amino Acid Change ^B^	Predicted Clinical Significance ^C^	
			PolyPhen-2 [95]	SIFT [96]
rs1305472211	3	A > T	Benign	Tolerated
rs1345615964	4	R > G	Benign	Deleterious
rs986654581	5	G > A	Benign	Tolerated
rs760771175	5	G > S	Benign	Tolerated
rs1240728249	10	Q > H	Possibly damaging	Deleterious
rs570487984	11	P > Q	Benign	Deleterious
rs570487984	11	P > R	Benign	Deleterious
rs764301435	11	P > S	Benign	Deleterious
rs1192097965	12	S > P	Benign	Tolerated
rs1397468624	13	T > I	Benign	Tolerated
rs1397468624	13	T > N	Benign	Deleterious
rs1385684307	19	P > L	Benign	Deleterious
rs1172122751	19	P > S	Benign	Deleterious
rs1455452771	21	A > V	Benign	Deleterious
rs934404193	22	P > Q	Benign	Deleterious
rs934404193	22	P > R	Benign	Deleterious
rs1316442498	22	P > S	Benign	Tolerated
rs1441432033	23	A > E	Benign	Tolerated
rs7968829	28	G > R	Probably damaging	Tolerated
rs1232428321	29	R > C	Benign	Deleterious
rs1414030211	29	R > P	Benign	Tolerated
rs914226102	31	R > C	Probably damaging	Deleterious
rs1356918678	38	E > K	Benign	Tolerated
rs759194451	39	P > A	Probably damaging	Tolerated
rs1457257561	41	G > D	Benign	Tolerated
rs1565699476	41	G > S	Benign	Tolerated
rs974285143	47	R > I	Probably damaging	Deleterious
rs762404803	48	P > R	Probably damaging	Deleterious
rs994288858	48	P > S	Probably damaging	Deleterious
rs1416276943	59	S > R	Possibly damaging	Deleterious
rs374963072	61	S > P	Probably damaging	Tolerated
rs192426102	68	A > G	Probably damaging	Deleterious
rs756682360	68	A > T	Probably damaging	Tolerated
rs79011113	69	E > K	Possibly damaging	Tolerated
rs745772022	70	A > T	Benign	Tolerated
rs1483054103	71	T > I	Benign	Tolerated
rs779988617	72	G > E	Probably damaging	Deleterious
rs768584493	73	E > D	Possibly damaging	Tolerated
rs1187672823	73	E > G	Probably damaging	Tolerated
rs1376518929	75	R > Q	Probably damaging	Tolerated
rs1433416843	82	K > R	Probably damaging	Deleterious
rs537621666	85	Q > H	Probably damaging	Tolerated
rs1176973487	87	V > A	Benign	Tolerated
rs1257572569	89	Q > K	Possibly damaging	Tolerated
rs151017786	92	P > A	Probably damaging	Tolerated
rs1251832694	92	P > R	Probably damaging	Tolerated
rs1384638234	93	A > D	Possibly damaging	Tolerated
rs1316622465	95	E > G	Probably damaging	Deleterious
rs755573046	95	E > Q	Probably damaging	Tolerated
rs1405500903	96	E > D	Benign	Tolerated
rs1272332080	99	E > G	Benign	Tolerated
rs1363332064	99	E > K	Benign	Tolerated
rs1302947401	101	S > F	Probably damaging	Tolerated
rs528046066	106	A > G	Probably damaging	Tolerated
rs745605177	107	E > K	Possibly damaging	Deleterious
rs745605177	107	E > Q	Probably damaging	Deleterious
rs1180385418	109	D > G	Probably damaging	Deleterious

^A^ Each amino acid position corresponds to the HMGA2 sequence shown in Figure 2B. ^B^ Amino acids are indicated by their single letter code. ^C^ Amino acid substitutions are deemed probably/possibly damaging or deleterious by PolyPhen-2 and SIFT, respectively.

**Table 2 cancers-15-01735-t002:** Non-coding RNAs that regulate *HMGA2* in CRC.

ncRNA	Effect	Reference
circRNA 100146	Promoter Inhibits miRNA 149, which inversely regulates *HMGA2* expression	[127]
miRNA 149	Inhibitor Suppresses *HMGA2* expression	[127]
LINC00963	Promoter Inhibits miRNA 532-3p, which inversely regulates *HMGA2* expression	[128]
miRNA 532-3p	Inhibitor Suppresses *HMGA2* expression	[128]
miRNA 330	Inhibitor Suppresses *HMGA2* expression	[129]
lncRNA PCAT6	Promoter Inhibits miRNA 204, which inversely regulates *HMGA2* expression Reduced sensitivity to 5-fluorouracil-based chemotherapies	[130]
miRNA 204	Inhibitor Negatively regulates *HMGA2* expression	[131]
circRNA NSUN2	Promoter Creates an RNA-protein complex by interacting with *IGF2BP2* and *HMGA2*	[132]
miRNA 543	Inhibitor Regulates *KRAS, MTA1* and *HMGA2* expression	[133]
miRNA 150	Inhibitor Downregulates *HMGA2* expression, which reduces Cyclin A levels, leading to cell cycle arrest	[134]
miRNA 4500	Inhibitor Suppresses *HMGA2* expression	[135]
miRNA 185-5p	Inhibitor Suppresses *HMGA2* expression	[136]
lncRNA DANCR	Promoter Acts as a molecular sponge for miRNA 185-5p, thus upregulating *HMGA2* expression	[136]
P53 inHibiting LncRNA (PiHL)	Promoter Reduces Oxaliplatin sensitivity	[137]
Let-7 miRNAs	Inhibitor Reduced Let-7 miRNAs abort tumor suppression by Let-7 targets (e.g., HMGA2)	[138]

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
