# Peer review of "Aberrant HMGA2 Expression Sustains Genome Instability That Promotes Metastasis and Therapeutic Resistance in Colorectal Cancer"

_cancers, 2023, doi:10.3390/cancers15061735_

Round 1

Reviewer 1 Report

This review focuses on the link between HMGA2 and genome instability-promoted metastasis and therapeutic resistance in colorectal cancer. It is an interesting and timely review with very useful information on this subject, which lays out the research landscape of HMGA2's complex contribution to metastasis and therapeutic resistance in colorectal cancer.

However, it is not very clear to readers what the specific mechanism is that explains the functional transition from maintaining genome integrity to sustaining genome instability. A brief discussion is necessary to help readers understand this issue. One way to address this is to cite the two-phased cancer evolution model. Genome instability is key to reorganizing new genomes that can survive drug treatment and promote metastasis in the macroevolutionary phase, while genome stability plays a role in helping cancer population growth in the microevolutionary phase. It is likely that HMGA2 can play different roles in different phases of cancer evolution.

There are two minor suggestions as well.

When discussing chromosomal instability, it is important to emphasize the structural alterations, including numerical and structural genome chaos such as massive translocation and polyploid giant cancer cells. Furthermore, it is important to mention cancer evolution, specifically the Big Bang model of colorectal tumor growth.

Overall, this review provides valuable insights into the link between HMGA2, genome instability, metastasis, and therapeutic resistance in colorectal cancer, and with these suggested improvements, it can help readers better understand the complex mechanisms underlying this link.

Author Response

The authors thank the reviewers for their helpful comments and valuable suggestions to improve this manuscript.

We have addressed the reviewer’s suggestions and our responses to the reviewer’s comments are listed below in italics.

Reviewer 1

This review focuses on the link between HMGA2 and genome instability-promoted metastasis and therapeutic resistance in colorectal cancer. It is an interesting and timely review with very useful information on this subject, which lays out the research landscape of HMGA2's complex contribution to metastasis and therapeutic resistance in colorectal cancer.

However, it is not very clear to readers what the specific mechanism is that explains the functional transition from maintaining genome integrity to sustaining genome instability. A brief discussion is necessary to help readers understand this issue. One way to address this is to cite the two-phased cancer evolution model. Genome instability is key to reorganizing new genomes that can survive drug treatment and promote metastasis in the macroevolutionary phase, while genome stability plays a role in helping cancer population growth in the microevolutionary phase. It is likely that HMGA2 can play different roles in different phases of cancer evolution.

The two-phased cancer evolution model and its importance for maintaining genome instability in CRC progression is explained in chapter 1.3.

We have referred to the potential role of HMGA2 in different phases of the 2 phased stochastic CRC progression model in the conclusion paragraph.

There are two minor suggestions as well.

When discussing chromosomal instability, it is important to emphasize the structural alterations, including numerical and structural genome chaos such as massive translocation and polyploid giant cancer cells. Furthermore, it is important to mention cancer evolution, specifically the Big Bang model of colorectal tumor growth.

We have added information on numerical (N-CIN) and structural (S-CIN) chromosome instability in chapter 1.3. The “Big Bang” model was introduced as it explains non-dominant heterogeneity that ultimately facilitates CRC metastasis and treatment resistance.

Overall, this review provides valuable insights into the link between HMGA2, genome instability, metastasis, and therapeutic resistance in colorectal cancer, and with these suggested improvements, it can help readers better understand the complex mechanisms underlying this link.

We thank the reviewer for these valuable suggestions to improving this manuscript.

Reviewer 2 Report

The manuscript reviews the impact of aberrant HMGA2 expression on genomic instability during colon carcinogenesis. The role of HMGA2 expression in metastasis and resistance to therapy is highlighted in the present manuscript. The work is important, however some points deserve attention:

1-    The hierarchy of topics is not adequate. The text is repetitive. Thus, I suggest that from topic 1.2 onwards, the manuscript follows this structure:

1.3 – “Genomic Instability in CRC” until page 5, line 182;

1.4 –  “HMGA2 and CRC” from page 5, line 183 to page 9, line 270 (including figure 2 and table 2, which is remunerated as table 1);

The topic 1.4 “Aberrant HMGA2 expression promotes CRC metastasis” turns into the topic 1.6;

1.5 “ The role of HMGA2 in genome maintenance”;

1.6 – “Aberrant HMGA2 expression promotes CRC metastasis” (incluinding table 1, which is remunerated as table 2);

1.7 – “HMGA2 promotes chemoresistance in CRC”.

2-    Page 3, line 133. Please delete “(discussed below).

3-    Figure 1. All abbreviations must be defined in the legend of the figure, such as MSI, CIN, etc.

4-    Page 6, lines 224-227. Please delete “In the following paragraphs, (…) in CRC”.

5-    Page 7, table 2. “Alteration” column. All changes presented in this column are missense. Would not it be better to delete this column and inform that all alterations are “missense” in the table title?

Author Response

The authors thank the reviewer for their helpful comments and valuable suggestions to improve this manuscript.

We have addressed the suggestions and our responses to the reviewer’s comments are listed below in italics.

Reviewer 2

Comments and Suggestions for Authors

The manuscript reviews the impact of aberrant HMGA2 expression on genomic instability during colon carcinogenesis. The role of HMGA2 expression in metastasis and resistance to therapy is highlighted in the present manuscript. The work is important, however some points deserve attention:

1-    The hierarchy of topics is not adequate. The text is repetitive. Thus, I suggest that from topic 1.2 onwards, the manuscript follows this structure:

1.3 – “Genomic Instability in CRC” until page 5, line 182;

1.4 –  “HMGA2 and CRC” from page 5, line 183 to page 9, line 270 (including figure 2 and table 2, which is remunerated as table 1);

The topic 1.4 “Aberrant HMGA2 expression promotes CRC metastasis” turns into the topic 1.6;

1.5 “ The role of HMGA2 in genome maintenance”;

1.6 – “Aberrant HMGA2 expression promotes CRC metastasis” (incluinding table 1, which is remunerated as table 2);

1.7 – “HMGA2 promotes chemoresistance in CRC”.

We have changed the sequence of the paragraphs to reflect the reviewer’s suggestion. 

2-    Page 3, line 133. Please delete “(discussed below).

This deletion was done 

3-    Figure 1. All abbreviations must be defined in the legend of the figure, such as MSI, CIN, etc.

The missing explanations for the abbreviations used in figure 1 were added to the legend  

4-    Page 6, lines 224-227. Please delete “In the following paragraphs, (…) in CRC”.

We deleted this sentence. 

5-    Page 7, table 2. “Alteration” column. All changes presented in this column are missense. Would not it be better to delete this column and inform that all alterations are “missense” in the table title?

With the restructuring of the manuscript, this table is now Table 1. The column “Alterations” was removed and we included in the table title that HMGA2 missense variants are listed.

We thank the reviewer for these helpful suggestions to improving this manuscript.

Round 2

Reviewer 2 Report

The authors improved the quality of the manuscript. The text is well structured and the figures are of good quality and have adequate legends.